# Beyond Single Tabs: A Transformative Few-Shot Approach to Multi-Tab Website Fingerprinting Attacks

## Abstract

Website Fingerprinting (WF) attacks allow passive eavesdroppers to deduce the websites a user visits by analyzing encrypted traffic, threatening user privacy. While current WF attacks achieve high accuracy, they typically assume single-tab browsing, which is unrealistic as users often open multiple tabs, creating mixed traffic. Existing multi-tab WF approaches require large datasets and frequent retraining due to evolving website content, limiting their practicality. In this paper, we introduce Few-shot Multi-tab Website Fingerprinting (FMWF), a novel approach designed to address the limitations of existing multi-tab WF attacks. FMWF directly tackles the challenges of mixed, overlapping traffic traces generated from multi-tab browsing, leveraging two key innovations: (1) an advanced data augmentation technique that synthesizes realistic multi-tab traffic sequences from easily collected single-tab traces, thereby dramatically reducing the need for large-scale real-world traffic data; and (2) a powerful fine-tuning algorithm based on transfer learning that adapts pre-trained models to new, multi-tab environments with minimal additional data. This two-stage framework enables FMWF to capture the complex effectively, overlapping traffic patterns inherent in multi-tab browsing while maintaining a high level of flexibility and significantly lowering computational and data collection burdens. Our experiments, conducted using real traffic traces collected from three widely-used browsers—Microsoft Edge, Google Chrome, and Tor Browser—highlight the superior performance of FMWF in both closed-world and open-world scenarios. Notably, FMWF achieves a minimum 12.3% improvement in accuracy compared to ARES (SP'23) [7], TMWF (CCS'23) [13], and BAPM (AC-SAC'21) [10] in the open-world scenario. The code with related datasets is available at https://anonymous.4open.science/r/FMWF-D164.

## CCS Concepts

• **Security and privacy** → **Privacy-preserving protocols**; • **Networks** → **Network privacy and anonymity**.

## Keywords

Multi-tab Website fingerprinting, Few-shot learning, Deep learning, Tor

**ACM Reference Format:**
Anonymous Author(s). 2025. Beyond Single Tabs: A Transformative Few-Shot Approach to Multi-Tab Website Fingerprinting Attacks. In *Proceedings of the ACM Web Conference 2025 (WWW '25), April 28–May 2, 2025, Sydney, Australia.* ACM, New York, NY, USA, 10 pages. https://doi.org/XXXXXXX.XXXXXXX

## 1 INTRODUCTION

As the internet continues to evolve and permeate daily activities such as online shopping, emailing, and social networking, browsers have become essential tools [12]. With this increased reliance, user concerns about privacy have also intensified. When a client accesses a website, routers along the communication path can observe or collect browsing data, potentially leading to privacy breaches for commercial exploitation or surveillance [17, 28, 33]. To safeguard privacy, users often obscure their online activity, with Tor [6] being the most popular anonymous network. Tor enhances privacy by rerouting data through multiple nodes, frequently changing IP addresses, and applying layered encryption to traffic, thereby thwarting traffic tracking efforts. Despite the privacy protections offered by networks like Tor, traffic analysis techniques, such as Website Fingerprinting (WF) attacks [3, 6, 23], can still deduce the websites that Tor users visit by scrutinizing traffic patterns, including packet size, transmission direction, and timing intervals.

WF attacks leverage Machine Learning (ML) and Deep Learning (DL) models to extract distinct traffic patterns from websites' traces, with state-of-the-art DL-based methods [1, 11, 18, 20, 21, 27] achieving outstanding accuracy, often exceeding 95%, in identifying the websites users visit. However, these attacks frequently operate under the single-page assumption, which presumes that users browse only one webpage per session. In reality, browsing habits often involve opening multiple tabs in rapid succession [8, 25], creating mixed and overlapping traffic patterns that are more difficult to distinguish. Research has shown that traditional WF attacks suffer from significant performance degradation in such multi-tab scenarios [14], which has spurred the development of new multi-tab WF techniques aimed at addressing this challenge.

Most existing multi-tab WF attacks [5, 29, 31, 32] adhere to a common architecture that segments browsing sessions into isolated traffic chunks associated with individual websites. However, this approach faces several critical drawbacks. First, these methods presuppose knowledge of the number of opened tabs, as exemplified by [31], which is designed for a fixed 2-tab setting. This assumption leads to significant performance degradation when the number of tabs is unknown or fluctuates dynamically—conditions that better reflect real-world browsing behaviors. Second, even under controlled conditions, these models are highly susceptible to WF defense mechanisms such as traffic padding and packet delays, both of which substantially reduce their accuracy [32]. Lastly, their performance worsens as the number of open tabs increases, making it more challenging to extract clean and distinct traffic chunks from

the increasingly mixed traffic streams. The noise introduced by overlapping traffic becomes more pronounced, ultimately resulting in a notable decline in identification accuracy.

In response to these challenges, more recent multi-tab WF attack methods [7, 10, 13] have been designed to address the limitations introduced by overlapping traffic traces specifically. These cutting-edge approaches, often employing Transformer-based architectures, achieve high website identification accuracy but at a high cost—they require vast amounts of multi-tab traffic data for training. The dynamic nature of websites further complicates this, as traffic patterns frequently change with content updates, making it difficult to keep these models up-to-date. Research [13, 20] has shown that models trained on traffic data collected just two weeks prior can experience a sharp drop in accuracy due to outdated patterns. This necessitates the constant collection of fresh, large-scale multi-tab traffic data, which is both costly and increases the risk of exposure. These limitations prompt an essential research question: **Can we devise a simplified, yet effective approach to conduct multi-tab website fingerprinting attacks that eliminates the need for prior knowledge of tab counts and large-scale training data?**

Therefore, in this paper, we introduce a novel approach to multi-tab website fingerprinting attacks that leverages transfer learning to overcome the limitations of existing methods. Unlike previous approaches, our method eliminates the need for complex traffic segmentation and extensive multi-tab traffic data collection, significantly reducing the bootstrapping time for WF attacks and enhancing feasibility in real-world scenarios. Our approach consists of two key stages:

(1) **Pre-training with Augmented Traffic Sequences:** We employ a carefully designed data augmentation method to synthesize a multi-tab dataset from single-tab traffic traces, enabling the model to build effective feature extractors and classifiers. By artificially combining traffic from individual websites into multi-tab traffic sequences, this stage reduces the dependency on large-scale real-world multi-tab datasets while still capturing essential traffic patterns for multi-tab environments.

(2) **Few-shot fine-tuning:** We adapt the pre-trained model from the previous stage to real-world conditions by fine-tuning it using a small amount of real-world multi-tab traffic data. This allows the model to further adjust to the overlapping and mixed nature of multi-tab traffic traces without requiring vast amounts of fresh data collection. Through transfer learning, this few-shot approach enables the model to achieve high accuracy and generalization in multi-tab WF attacks, even in dynamic or changing traffic environments.

This two-stage framework not only enables our model to effectively handle mixed and overlapping traffic from multiple tabs but also eliminates the need for prior knowledge of the number of open tabs, a common assumption in previous studies.

The main contributions of this paper are as follows:

- To the best of our knowledge, we propose the first few-shot multi-tab WF attack method named FMWF. By utilizing a novel data augmentation method specifically designed for multi-tab website fingerprinting attacks, FMWF removes the need for a large number of real-world multi-tab traffic traces.

- In the few-shot scenario, we move away from using large models, such as Transformers, which are typically resource-intensive and data-hungry. Instead, we implement lightweight neural networks as feature extractors and classifiers, which, through a well-crafted fine-tuning transfer learning algorithm, are capable of achieving competitive performance. This approach significantly reduces the model complexity while maintaining accuracy, making it more efficient and practical for real-world applications where data and computational resources are limited.

- To demonstrate the robust performance of our model, we conduct comprehensive tests on Tor, Microsoft Edge, and Google Chrome datasets. On the Tor dataset, it achieves a top-k accuracy of 0.910 in the 5-tab setting using just 25-shot learning samples per label. On the Microsoft Edge and Google Chrome datasets, which include rich features such as packet sizes, FMWF reaches a top-k accuracy of 0.885 in the 5-tab setting with only 3-shot learning samples per label.

- In the more challenging open-world scenario, we observe a remarkable minimum improvement of 12.3% compared to ARES (SP'23) [7], TMWF (CCS'23) [13], and BAPM (AC-SAC'21) [10] when testing on non-monitored websites.

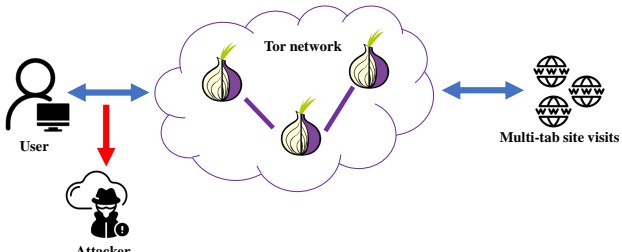

**Figure 1: Threat model of FMWF. Users can simultaneously open multiple tabs to visit different websites, and an attacker can eavesdrop on the encrypted traffic within the Tor network.**

## 2 BACKGROUND

### 2.1 Threat Model

Our threat model involves a passive adversary capable of observing encrypted network traffic, similar to previous WF attack studies [7, 10, 12, 13, 21, 30]. As shown in Figure 1, users access websites through anonymous systems like Tor, which safeguard privacy via multi-layer encryption and relay nodes. The attacker, potentially an ISP or a device between the ISP and the first Tor node, passively monitors traffic without altering it, unlike active attackers who can delay or drop packets. Though unable to modify the traffic, the adversary observes encrypted packet metadata (size, direction, timing) to infer the websites visited by the user. Using machine learning models trained on traffic patterns, the attacker can associate observed traffic with specific websites, even in multi-tab browsing

scenarios where traffic from multiple websites may intermingle within a session.

## 2.2 Single-tab WF Attacks

**Deep-learning-based WF:** Deep learning was first applied to website fingerprinting attacks on Tor by Abe *et al.* [1], who utilized stacked auto-encoders, achieving results comparable to traditional feature extraction methods on simple direction sequences. This marked a shift from manually crafted features to automated feature extraction, with the performance of deep learning models improving as more complex network structures were introduced.

Rimmer *et al.* [20] demonstrated that deep learning models are robust against changes in website content, showcasing the feasibility of automated feature learning in WF attacks. Building on this, Sirinam *et al.* [21] proposed the convolutional neural network (CNN) model Deep Fingerprinting (DF), specifically designed for WF attacks. DF improved the model's ability to capture subtle patterns in encrypted traffic by incorporating multiple advanced convolutional blocks, achieving an accuracy of 98.3% in a closed-world scenario and establishing CNNs as powerful tools for WF attacks. Following this, Bhat *et al.* [2] introduced Var-CNN, a framework that combines semi-automated feature extraction and demonstrated better performance with smaller datasets, addressing the common bottleneck of deep learning models that typically require large amounts of labeled data.

**Transfer-learning-based WF:** Supervised learning methods [4, 34] often require extensive labeled data, which is time-consuming and costly. Transfer learning [24, 35] provides a solution by leveraging previously learned knowledge for new tasks. In recent WF attacks [22, 26, 30], transfer learning has been applied in few-shot learning scenarios. This typically involves pre-training on a dataset with abundant labeled data, fine-tuning with limited samples, and testing.

Recent works, such as Triplet Fingerprinting (TF) [22], Adaptive Fingerprinting (AF) [26], and Contrastive Fingerprinting (CF) [30], aim to refine feature extraction for few-shot scenarios. CF integrates contrastive learning and data augmentation for a more efficient WF attack, improving representation quality and training efficiency. TF enhances feature distinction through a triplet network, while AF employs adversarial domain adaptation to learn domain-invariant features. However, the effectiveness of these WF attacks relies on the strong assumption that only a single website is visited during a browsing session, a scenario that does not always hold as users often open multiple websites simultaneously or within a short period [9, 14, 29, 31].

## 2.3 Multi-tab WF Attacks

Recent studies have begun to address the challenges associated with multi-tab website fingerprinting (WF) attacks, aiming to improve website identification accuracy in complex mixed-traffic scenarios. Notably, models such as Transformer-based Multi-tab Website Fingerprinting [7, 10, 13] have introduced innovative strategies to effectively handle overlapping and mixed traffic segments. These approaches frame multi-tab browsing as either a multi-label classification task or a sequence prediction problem, utilizing advanced neural network architectures like transformers. By doing so, these

models can extract both local and global traffic patterns from the data, thereby enhancing the accuracy of website identification in scenarios where multiple tabs are open simultaneously.

These advancements represent a significant step forward in the field of WF, yet they also reveal inherent challenges. The reliance on large amounts of labeled data and the assumption of static tab counts in existing models limit their practical application, particularly in dynamic real-world environments. As user behavior becomes increasingly complex, the need for robust and adaptable models that can generalize across varying traffic conditions and tab configurations becomes paramount.

We propose the Few-shot Multi-tab Website Fingerprinting attack (FMWF), which leverages transfer learning and data augmentation to achieve high identification accuracy with minimal training data. This eliminates the need for extensive data collection and constant retraining. FMWF handles complex, overlapping traffic traces without relying on prior knowledge of tab counts, making it highly flexible and practical in real-world settings, while also reducing computational demands and data requirements typically seen in multi-tab WF models.

## 3 METHOD

In this section, we present the detailed structure and the workflow of FMWF. Before delving into the various components of FMWF, we first provide an overview of the framework.

## 3.1 Overview

Previous Transformer-based multi-tab WF attacks have achieved high website identification accuracy by utilizing extensive multi-tab traffic traces. However, the characteristics of website traffic can fluctuate due to variations in website content and user network conditions. Maintaining high attack accuracy often necessitates the frequent collection of substantial amounts of new traffic traces to retrain the attack model. This process can be both costly and impractical, posing significant challenges to the real-world applicability of website fingerprinting attacks.

To address the limitations of existing multi-tab WF attacks, we propose FMWF, a novel multi-tab website fingerprinting attack method based on transfer learning that is effective in few-shot scenarios. As illustrated in Figure 2, the FMWF attack consists of two main stages with a testing phase:

(1) **Pre-training with Augmented Traffic Sequences**: In the pre-training stage, we introduce a novel data augmentation method to simulate the behavior of users accessing multiple websites simultaneously. This method generates synthetic multi-tab browsing sessions by combining individual traffic patterns from various websites, allowing FMWF to learn critical features related to multi-tab traffic traces. The pre-training stage leverages the knowledge obtained from synthetic data to enhance the model's ability to recognize and classify real-world multi-tab traffic efficiently.

(2) **Few-shot Fine-tuning for Real-world Adaptation**: After pre-training, the model enters the fine-tuning stage, adapting to real-world multi-tab traffic using few-shot labeled data. This fine-tuning process refines the learned

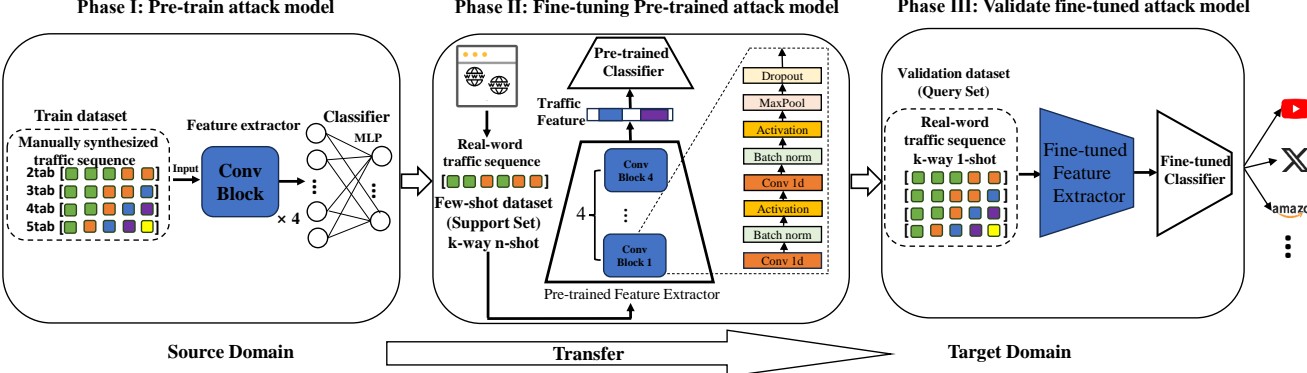

**Figure 2: Our lightweight WF attack, FMWF, consists of two key components: a feature extractor and a classifier, both built using lightweight neural networks. These components work in tandem with a fine-tuning algorithm derived from transfer learning, enabling efficient and accurate multi-tab website fingerprinting with minimal training data.**

features, adjusting the model to handle real-world complex, mixed traffic patterns typical of multi-tab browsing sessions. By focusing on a few-shot learning method, this stage enables the model to generalize to real-world multi-tab quickly and dynamically changing traffic environments, without requiring continuous large-scale retraining.

(3) **Testing on Dynamic Traffic**: Once fine-tuned, FMWF is evaluated on diverse datasets to assess its performance in close and open-world scenarios.

## 3.2 Pre-training with Augmented Traffic Sequences

In the pre-training stage of the attack model, three main components are involved: the pre-training dataset, the feature extractor, and the classifier.

**Pre-training dataset.** For the pre-training dataset, we propose a data augmentation method. Instead of collecting multi-tab traffic traces directly from the real world, we manually synthesize a multi-tab dataset by combining single-tab traffic sequences. This method achieves data augmentation and mitigates the challenges associated with collecting multi-tab datasets.

Collecting multi-tab datasets is more challenging than single-tab datasets due to the exponential increase in the size of the label set. For example, monitoring one hundred websites generates over four thousand labels for two-tab traffic, making it difficult for an attacker to gather and frequently update a large-scale dataset. However, one can easily generate large-scale datasets by manually synthesizing multi-tab traffic sequences. This data augmentation strategy enables the pre-training of attack models on multi-tab traffic without the need to collect real-world multi-tab datasets.

In [29], multi-tab traffic traces are categorized into three fundamental types based on their time interval relationships: positive, zero, and negative. Figure 3 illustrates these scenarios for 2-tab traffic traces. As shown in Figure 4, to ensure a complete characterization of each website's traffic, we manually synthesize multi-tab traffic traces specifically using the zero time interval type. This strategy allows the feature extractor to more effectively learn the

traffic characteristics of each website, leading to a more robust pre-trained attack model.

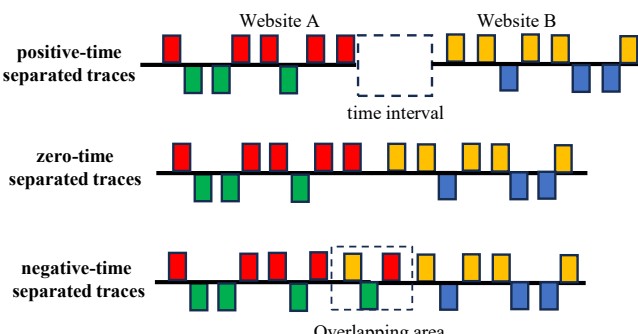

**Figure 3: Three basic situations of 2-tab traffic traces. Depending on whether a user accesses Site B via a new tab before Site A finishes loading, 2-tab traffic patterns can be categorized into these three types.**

Specifically, let $(t, y)$ represent a traffic instance generated by a user browsing websites, where $t$ is the traffic sequence consisting of $n$ packets, and $y$ is the label indicator vector for that instance. We use multi-hot encoding to represent the label vector, denoted as $y = [y_0, y_1, \ldots, y_{99}]$. If the user visits the $i$-th and $j$-th websites among the 100 monitored websites, $y_i$ and $y_j$ are set to 1, while all other label positions are set to 0. In the pre-training dataset, we concatenate the single-tab traffic sequences generated by a user visiting websites $i$ and $j$ separately to form a multi-tab traffic sequence, $\tilde{t} = Aug(t_i, t_j)$. This allows us to obtain a manually synthesized instance $(\tilde{t}, \tilde{y})$, where the augmented instance shares the same label as a real-world instance, i.e., $\tilde{y} = y$.

**Feature extractor and Classifier.** The FMWF attack model employs a specialized feature extractor and classifier to tackle critical challenges in multi-tab website fingerprinting. The feature extractor is built upon a convolutional neural network (CNN) architecture that effectively addresses noise reduction from overlapping traffic segments [21], [19], adapts to dynamic website traffic patterns,

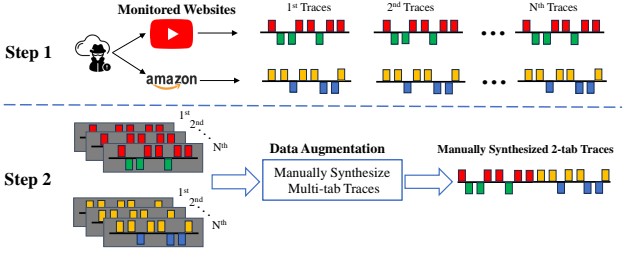

**Figure 4: Manually Synthesized Multi-Label Traffic Traces from sing-table sessions for the Pre-Training Stage.**

and maintains flexibility in processing variable-length sequences. Key techniques such as batch normalization and dropout are utilized to enhance the robustness of feature representations and prevent overfitting, ensuring that the model can accurately capture the relevant characteristics of user browsing behavior even when the number of open tabs varies.

The classifier, a multi-layer perceptron (MLP), operates on the feature embeddings generated by the extractor to predict the labels of visited websites. It incorporates a weighted loss function to mitigate data imbalance in few-shot learning scenarios, ensuring that underrepresented classes are adequately represented in the model's predictions. Designed for real-time processing efficiency, the classifier utilizes fully connected layers combined with batch normalization and ReLU activations, concluding with a softmax layer for probability distribution over potential labels. Together, these components of the FMWF model enhance its accuracy and robustness and can be quickly adapted to real-world datasets during the fine-tuning phase of the few-shot classification task.

## 3.3 Few-shot Fine-tuning for Real-world Adaptation

During the fine-tuning stage of the FMWF model, we leverage a support set derived from real-world multi-tab traffic to enhance the feature extractor and classifier. The primary aim of this process is to improve the model's accuracy in identifying labels corresponding to actual user traffic patterns. As highlighted in previous research [16], fine-tuning algorithms have demonstrated significant efficacy in transfer learning, allowing models to adapt quickly to new data.

One of the key challenges in feature extraction for multi-tab traffic is the blending of traffic from various websites, which introduces substantial noise that can obscure the unique characteristics of individual sites. To address this, we initiate fine-tuning by feeding a limited number of real-world multi-tab traffic sequences into the pre-trained feature extractor. This generates robust feature embeddings that encapsulate the traffic patterns of each website. These embeddings serve as input for the multi-layer perceptron (MLP) classifier, which is then fine-tuned to enhance its predictive performance before being tested with the query set. This two-step process ensures that the FMWF model remains adaptable and effective, even in the dynamic environment of real-world web traffic.

## 3.4 Testing on Dynamic Traffic

In the testing phase of the Few-shot Multi-tab Website Fingerprinting (FMWF) model, we utilize a query set designed for a k-way 1-shot scenario, meaning it consists of k distinct labels, each represented by only a single sample. The testing process involves analyzing unlabeled multi-tab traffic traces captured from anonymous networks, such as Tor, as well as from mainstream browsers like Microsoft Edge and Google Chrome.

## 3.5 Algorithm Summary

FMWF presents a groundbreaking approach to multi-tab website fingerprinting by utilizing transfer learning and few-shot fine-tuning. This method effectively overcomes the limitations of traditional WF techniques that either assume single-tab browsing or require extensive collections of real-world multi-tab traffic traces. Instead of relying on large-scale datasets and frequent retraining, FMWF synthesizes multi-tab traffic sequences through data augmentation during the pre-training phase, enabling the model to capture essential traffic features from artificially combined single-tab traces.

The architecture of FMWF incorporates a streamlined design, utilizing a CNN as the feature extractor and an MLP for classification. This innovative integration of synthetic data generation and few-shot fine-tuning enhances the model's efficiency and adaptability in dynamic real-world scenarios. By establishing a robust and practical solution for multi-tab WF attacks, FMWF demonstrates significant advancements in accuracy and operational feasibility, making it a valuable contribution to website fingerprinting.

## 4 EXPERIMENTAL EVALUATIONS

In this section, we conduct comprehensive experiments on real-world multi-tab datasets to address the following research questions regarding FMWF:

- **RQ I**: How does the number of fine-tuning traces $N$ influence top-K accuracy (A@K), Precision, Recall, and F1-Score across different tab settings in few-shot multi-tab scenarios compared to state-of-the-art models?
- **RQ II**: How effective is FMWF in real-world scenarios without prior knowledge of the number of opened tabs?
- **RQ III**: How does the number of labels ($M$) during the few-shot fine-tuning stage influence the generalization performance of FMWF in multi-tab classification tasks?
- **RQ IV**: How does FMWF perform on diverse browser datasets (Google Chrome and Microsoft Edge), especially with limited fine-tuning samples per label?
- **RQ V**: How effective is FMWF in the open-world scenario?

## 4.1 Experiment Setup

*4.1.1 Dataset.* In this study, we collected three datasets by using three major browsers to access target websites and capture traffic data: the Tor browser, Microsoft Edge, and Google Chrome.

**Tor Dataset.** Using the method outlined in [7], we collected Tor traffic data over two months, leveraging 15 Alibaba Cloud servers with varying hardware configurations, located across the US and Singapore. This setup allowed us to simulate different user scenarios, incorporating variations in device performance, network

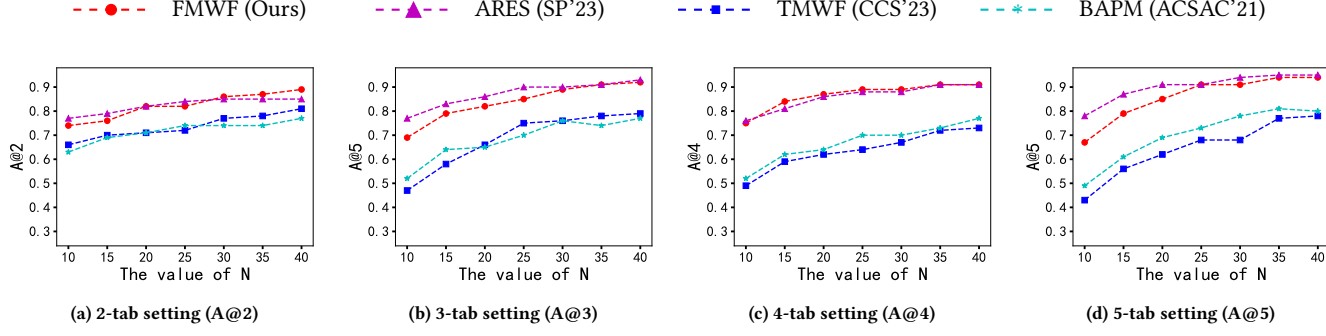

**Figure 5: Comparison of FMWF with state-of-the-art multi-tab WF attacks, where A@K represents the top-k Accuracy.**

conditions, and periods. Based on the website traffic rankings[1], we selected the top 100 most popular URLs. For each site, we collected 100 single-tab traffic samples by repeatedly accessing these websites via the Tor Browser. To collect multi-tab datasets, we randomly combined these 100 websites to generate 2-tab, 3-tab, 4-tab, and 5-tab scenarios, also using the Tor Browser. In the open-world scenario, we used the Tor open-world dataset provided by [7].

**Microsoft Edge & Google Chrome Dataset.** We adopted a dataset collection method similar to previous research [12]. Traffic data was captured using Tshark, while Python and PyAutoGUI were employed for automation. Using these two major browsers, we collected traffic data by repeatedly accessing 100 monitored websites [2], capturing 50 traffic samples per website. We randomly combined the 100 target websites to collect 2-tab, 3-tab, 4-tab, and 5-tab datasets.

*4.1.2 Metrics.* To evaluate the performance of our multi-tab WF attack, we adopt the metrics commonly used in related studies [7, 15]. In the closed-world scenario, multi-tab WF attacks are framed as a multi-tab multi-class classification problem. For this, we employ the multi-tab-specific metric, Top-k Accuracy (A@K), and traditional multi-class metrics such as Precision, Recall, and F1-score. These metrics comprehensively evaluate the model's performance by assessing the prediction outcomes for each website across all instances.

In the open-world scenario, we assess the proposed method by following established practices from previous studies [20, 21]. In this context, all non-monitored websites are aggregated into a single category, enabling us to evaluate the model's performance in accurately identifying the websites a user visits, even amidst a vast array of potentially unseen sites. This approach not only tests the adaptability and robustness of FMWF but also reflects more realistic browsing conditions, where users interact with a mix of known and unknown websites. We present the details of the metrics in Appendix A.

## 4.2 Closed-world Evaluation

In the closed-world scenario, we selected the current state-of-the-art end-to-end multi-tab website fingerprinting attack methods, ARES (SP'23) [7], TMWF (CCS'23) [13], and BAPM (ACSAC'21) [10], as comparative methods for evaluating the FMWF attack model in the

following experiments. To assess the robustness of FMWF, we tested it using datasets collected from three mainstream browsers—Google Chrome, Microsoft Edge, and Tor Browser. We used 5 different validation sets for each experiment and calculated the average as the final result. Except for Experiment AQ IV, all other experiments were conducted using the Tor dataset.

**AQ I: Evaluating the Influence of $N$ on Top-K Accuracy (A@K) Performance.** In this experiment, we investigate the impact of varying the number of fine-tuning traces ($N$) per label on the performance of our multi-tab WF attack model across different tab settings during the few-shot fine-tuning stage. We conduct experiments using a 100-way $N$-shot fine-tuning dataset, where $N$=10,15,20,25,30,35,40. The primary evaluation metric is A@K, which is widely used in similar studies [7, 15], to assess the model's generalization capability in multi-tab browsing scenarios. A@K measures the model's ability to correctly identify the actual visited websites within the top K predictions.

**Benefits from the fine-tuning algorithm, our lightweight WF attack model demonstrates strong competitiveness against current state-of-the-art Transformer-based large WF attack models.** As illustrated in Figure 5, FMWF consistently surpasses the advanced models (TMWF and BAPM) across all tab settings. For instance, in the 2-tab scenario, when $N$=20, the A@2 score of FMWF exceeds 0.8, significantly outperforming TMWF and BAPM, whose A@2 scores remain below 0.70. This trend continues in the 3-tab and 4-tab settings, where FMWF sustains a clear advantage in top-k accuracy. While ARES, a state-of-the-art multi-tab WF attack model based on a sophisticated transformer architecture, performs closely to FMWF due to its ability to capture traffic features from pre-trained datasets, FMWF remains competitive with its lightweight model structure. As $k$ increases, A@K scores remain consistently high, showcasing FMWF's robustness in handling complex multi-tab scenarios, all while leveraging few-shot learning to maintain strong identification accuracy even in the most challenging multi-tab environments.

**AQ I: Impact of $N$, the Number of Fine-Tuning Traces on Precision, Recall, and F1-Score Across Varying Numbers of Tabs in the Few-Shot Fine-Tuning Stage.** As shown in Table 1, FMWF consistently surpasses other state-of-the-art multi-tab WF attacks across most configurations in terms of Precision, Recall, and F1-Score. For instance, in the 100-way 40-shot setting, FMWF attains an impressive F1-Score of 0.904 in the 2-tab scenario, substantially outperforming BAPM and TMWF, which achieve F1-Scores of 0.744

---
[1]https://majestic.com/reports/majestic-million
[2]https://top.chinaz.com/all/

**Table 1: Comparison of FMWF with prior arts in Precision, Recall, and F1-Score across different tabs and sample numbers (N).**

| Method | N | 2-tab | | | 3-tab | | | 4-tab | | | 5-tab | | |
|---|---|---|---|---|---|---|---|---|---|---|---|---|---|
| | | Precision | Recall | F1-score | Precision | Recall | F1-score | Precision | Recall | F1-score | Precision | Recall | F1-score |
| BAPM (ACSAC'21) | 10 | 0.605 | 0.730 | 0.634 | 0.571 | 0.647 | 0.587 | 0.538 | 0.668 | 0.582 | 0.431 | 0.560 | 0.476 |
| | 20 | 0.691 | 0.805 | 0.727 | 0.688 | 0.760 | 0.711 | 0.648 | 0.770 | 0.692 | 0.634 | 0.686 | 0.645 |
| | 30 | 0.734 | 0.800 | 0.746 | 0.749 | 0.790 | 0.746 | 0.695 | 0.808 | 0.737 | 0.679 | 0.760 | 0.710 |
| | 40 | 0.738 | 0.825 | 0.761 | 0.749 | 0.820 | 0.774 | 0.736 | 0.793 | 0.746 | 0.797 | 0.848 | 0.813 |
| TMWF (CCS'23) | 10 | 0.676 | 0.725 | 0.684 | 0.528 | 0.493 | 0.573 | 0.471 | 0.493 | 0.469 | 0.605 | 0.656 | 0.623 |
| | 20 | 0.783 | 0.775 | 0.770 | 0.647 | 0.717 | 0.665 | 0.612 | 0.623 | 0.597 | 0.702 | 0.768 | 0.726 |
| | 30 | 0.820 | 0.835 | 0.818 | 0.708 | 0.780 | 0.729 | 0.687 | 0.760 | 0.710 | 0.806 | 0.862 | 0.826 |
| | 40 | 0.838 | 0.860 | 0.844 | 0.706 | 0.763 | 0.714 | 0.702 | 0.818 | 0.747 | 0.830 | 0.880 | 0.849 |
| AERS (SP'23) | 10 | 0.857 | **0.805** | **0.812** | 0.776 | 0.720 | **0.727** | 0.855 | **0.793** | **0.811** | 0.837 | **0.778** | **0.790** |
| | 20 | 0.865 | 0.855 | 0.847 | **0.873** | 0.827 | 0.832 | 0.909 | 0.883 | 0.890 | 0.926 | 0.890 | **0.902** |
| | 30 | 0.887 | 0.885 | **0.880** | 0.898 | 0.877 | **0.877** | 0.928 | 0.903 | **0.911** | 0.928 | 0.904 | 0.908 |
| | 40 | **0.903** | 0.880 | 0.884 | **0.935** | 0.897 | 0.905 | **0.934** | 0.908 | 0.912 | 0.921 | 0.918 | **0.918** |
| **FMWF (Ours)** | 10 | **0.885** | 0.800 | 0.794 | **0.779** | **0.725** | 0.724 | **0.890** | 0.770 | 0.775 | **0.849** | 0.740 | 0.757 |
| | 20 | **0.883** | **0.865** | **0.849** | 0.867 | **0.840** | **0.835** | **0.930** | **0.893** | **0.889** | **0.929** | **0.892** | 0.894 |
| | 30 | **0.894** | **0.896** | 0.879 | 0.893 | **0.881** | 0.875 | **0.935** | **0.918** | 0.909 | 0.927 | **0.908** | **0.912** |
| | 40 | 0.916 | **0.920** | **0.904** | 0.902 | **0.926** | **0.909** | 0.928 | **0.920** | **0.914** | **0.925** | **0.920** | 0.915 |

and 0.751, respectively. This trend continues in the 3-tab and 4-tab settings, where FMWF maintains superior and stable results.

Additionally, FMWF demonstrates exceptional stability as the number of shots increases, ensuring consistent performance across various scenarios. For example, even with just 30 fine-tuning traces per label in the 5-tab scenario, FMWF achieves an F1-Score of 0.912, outperforming ARES (0.908), BAPM (0.710), and TMWF (0.826). This consistency is observed across all tab settings and shot numbers, underscoring FMWF's robustness in navigating complex multi-tab environments with limited fine-tuning data.

**Table 2: Performance of FMWF without prior knowledge of the number of tabs, across sample numbers (N).**

| The value of N | Precision | Recall | F1-score |
|---|---|---|---|
| 10 | 0.820 | 0.660 | 0.675 |
| 15 | 0.849 | 0.758 | 0.764 |
| 20 | 0.872 | 0.818 | 0.811 |
| 25 | 0.877 | 0.849 | 0.835 |
| 30 | 0.882 | 0.866 | 0.855 |
| 35 | 0.890 | 0.882 | 0.870 |
| 40 | 0.905 | 0.907 | 0.890 |

**AQ II: Evaluating the Performance of FMWF Without Prior Knowledge of the Number of Tabs.** We further evaluate the performance of FMWF in a scenario where the number of opened tabs is unknown to the adversary. We tested using a dataset with an equal proportion mixture of 2-tab, 3-tab, 4-tab, and 5-tab traffic traces. This dynamic setting is more practical and challenging as it simulates real-world browsing behavior where users may open multiple tabs simultaneously without revealing the exact number of tabs. To assess the performance of our model under these conditions, we use Precision, Recall, and F1-score metrics, as shown in Table 2. FMWF demonstrates strong performance across all values of $N$. For instance, with $N$=25, it achieves a Precision of 0.877, Recall of 0.849, and F1-score of 0.835. Notably, the model exhibits

consistent and robust performance as $N$ increases, indicating that FMWF can effectively handle multi-tab traffic even without prior knowledge of the number of tabs. This showcases the adaptability and effectiveness of FMWF in real-world browsing environments.

**AQ III: Impact of $M$, the Number of Labels in the Few-Shot Fine-Tuning Stage.** For this experiment, we collected 2-tab traffic traces corresponding to 400 labels using Tor browser versions V12.5.6 and V13.5. We then evaluated the impact of $M$ (the number of labels) during the few-shot fine-tuning stage, alongside assessing the robustness of FMWF across different browser versions. The model is tested across different settings, with $M$=200,300,400 and $N$=10,20,30,40 (representing the number of samples per label). As shown in Table 3, increasing the number of labels ($M$) consistently results in strong performance. For example, in the 400-way task, the model achieves a top-2 accuracy of 0.883 and an F1-score of 0.887. This demonstrates that increasing the number of labels ($M$) during fine-tuning allows FMWF to maintain its ability to generalize effectively in complex, multi-tab classification tasks.

**AQ IV: Performance of FMWF on Google Chrome and Microsoft Edge Datasets.** In this experiment, we evaluate the effectiveness of FMWF on two widely used but less explored non-anonymous browsers, Google Chrome and Microsoft Edge, which capture a richer set of network features such as packet sizes and directions. In contrast to the anonymous Tor browser, FMWF requires significantly fewer samples to achieve high accuracy. We compare FMWF against three state-of-the-art models (ARES, TMWF, and BAPM) across 2-tab, 3-tab, 4-tab, and 5-tab scenarios, utilizing varying numbers of samples per label ($N$).

As shown in Table 4, FMWF consistently outperforms the other models in terms of top-k accuracy across most tab settings. For example, in the 2-tab scenario, with just 3 samples per label, FMWF achieves a remarkable top-2 accuracy score of 0.964 on Google Chrome and 0.972 on Microsoft Edge when $N$=3, surpassing ARES (0.960 on Chrome and 0.958 on Edge), TMWF (0.637 on Chrome and 0.794 on Edge), and BAPM (0.783 on Chrome and 0.822 on

**Table 3: Performance Evaluation Across Different Label Quantities (M) in Few-Shot Fine-Tuning.**

| The value of N | 200-way | | | | 300-way | | | | 400-way | | | |
|---|---|---|---|---|---|---|---|---|---|---|---|---|
| | A@2 | Precision | Recall | F1-score | A@2 | Precision | Recall | F1-score | A@2 | Precision | Recall | F1-score |
| 10 | 0.732 | 0.805 | 0.826 | 0.818 | 0.711 | 0.770 | 0.800 | 0.767 | 0.645 | 0.779 | 0.753 | 0.744 |
| 20 | 0.838 | 0.874 | 0.905 | 0.878 | 0.822 | 0.859 | 0.882 | 0.855 | 0.804 | 0.836 | 0.849 | 0.828 |
| 30 | 0.890 | 0.911 | 0.937 | 0.918 | 0.873 | 0.889 | 0.922 | 0.896 | 0.815 | 0.861 | 0.893 | 0.863 |
| 40 | 0.914 | 0.929 | 0.955 | 0.934 | 0.901 | 0.906 | 0.940 | 0.916 | 0.883 | 0.887 | 0.909 | 0.887 |

**Table 4: Comparison of FMWF with prior arts in A@K across different tabs and sample numbers (N).**

| Method | The value of N | 2-tab (A@2) | | 3-tab (A@3) | | 4-tab (A@4) | | 5-tab (A@5) | |
|---|---|---|---|---|---|---|---|---|---|
| | | Google | Microsoft | Google | Microsoft | Google | Microsoft | Google | Microsoft |
| BAPM (ACSAC'21) | 1 | 0.586 | 0.638 | 0.491 | 0.573 | 0.457 | 0.442 | 0.338 | 0.352 |
| | 2 | 0.744 | 0.783 | 0.654 | 0.726 | 0.635 | 0.617 | 0.525 | 0.518 |
| | 3 | 0.783 | 0.822 | 0.712 | 0.779 | 0.695 | 0.663 | 0.603 | 0.635 |
| TMWF (CCS'23) | 1 | 0.356 | 0.491 | 0.221 | 0.352 | 0.184 | 0.247 | 0.124 | 0.208 |
| | 2 | 0.542 | 0.706 | 0.484 | 0.606 | 0.408 | 0.534 | 0.318 | 0.410 |
| | 3 | 0.637 | 0.794 | 0.613 | 0.674 | 0.583 | 0.639 | 0.467 | 0.563 |
| AERS (SP'23) | 1 | **0.816** | 0.818 | **0.871** | 0.690 | 0.706 | 0.658 | **0.607** | 0.601 |
| | 2 | 0.924 | **0.944** | 0.900 | **0.892** | 0.821 | **0.870** | **0.830** | 0.822 |
| | 3 | 0.960 | 0.958 | **0.922** | 0.910 | **0.903** | 0.901 | 0.852 | **0.890** |
| **FMWF (Ours)** | 1 | 0.782 | **0.826** | 0.732 | **0.694** | **0.715** | **0.672** | 0.588 | **0.614** |
| | 2 | **0.933** | 0.940 | **0.904** | 0.857 | **0.840** | 0.868 | 0.824 | **0.841** |
| | 3 | **0.964** | **0.972** | 0.915 | **0.913** | 0.895 | **0.904** | **0.867** | 0.885 |

Edge). This strong performance persists in more complex multi-tab scenarios, where FMWF continues to deliver high A@K scores, highlighting its robustness and adaptability across different browser environments.

**Table 5: Performance Evaluation of FMWF in the Open-World Scenario with Different Numbers of Traces (N) for Few-Shot Fine-Tuning.**

| Method | Number of Traces (N) | | | | |
|---|---|---|---|---|---|
| | N=1 | N=3 | N=5 | N=7 | N=10 |
| BAPM (ACSAC'21) | 0.568 | 0.685 | 0.703 | 0.725 | 0.738 |
| TMWF (CCS'23) | 0.355 | 0.579 | 0.657 | 0.708 | 0.855 |
| ARES (SP'23) | 0.364 | 0.485 | 0.533 | 0.578 | 0.655 |
| **FMWF(Ours)** | **0.925** | **0.943** | **0.949** | **0.954** | **0.960** |

## 4.3 AQ V: Open-world Evaluation

In the open-world scenario, FMWF is assessed on its ability to differentiate between monitored and non-monitored websites. As in previous studies [7, 10, 13], all non-monitored websites are treated as a single broad category, while each monitored website is considered a distinct class. To mitigate data imbalance, we adopt the same strategy as earlier works [7, 30] by combining closed-world and open-world instances under the same tab settings. For instance, the 2-tab closed and open-world instances are merged to conduct the 2-tab open-world experiment.

We use $N$ = 1, 3, 5, 7, 10 to represent the number of traces per monitored website in the support set for few-shot fine-tuning. In the query set, we test the model with 100 monitored and 1000 unmonitored traces. As shown in Table 5, FMWF consistently achieves over 90% top-2 accuracy across all settings. **FMWF exhibits competitive performances against transformer-based large models, further highlighting its efficiency and robustness in more complex open-world scenarios.**

## 5 Conclusion and Future Work

In this work, we have introduced FMWF, a novel few-shot multi-tab website fingerprinting (WF) attack method that leverages transfer learning to address the limitations of existing WF approaches. Unlike traditional methods that depend on extensive datasets and prior knowledge of the number of open tabs, FMWF employs a two-stage framework that integrates data augmentation and few-shot fine-tuning. This design enables the efficient handling of multi-tab traffic traces while requiring minimal real-world data, making it more adaptable to realistic browsing behaviors.

Looking ahead, we plan to investigate potential defense mechanisms against multi-tab WF attacks and explore how our method can be enhanced to counteract these defenses. Common strategies in website fingerprinting include traffic obfuscation techniques, such as packet padding, traffic morphing, and random delays, which aim to obscure traffic traces and make them less distinguishable. Given that multi-tab traffic is already challenging to analyze due to inherent overlaps, incorporating defenses will introduce an additional layer of complexity, presenting a valuable avenue for future research in enhancing both attack and defense methodologies in this domain.

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

# A  Detailed Settings for Experiment

## A.1  Parameter Settings

We conducted an extensive search of the hyperparameter space to evaluate and select the optimal hyperparameters for TMWF. Table 6 lists the key hyperparameters we tuned, the range of candidate values, and the final selected values.

**Table 6: Hyperparameter selection for the FMWF**

| Parameters | Search Space | Selected Value |
|---|---|---|
| Feature Extractor | GoogleNet, ResNet, DF | DF |
| Classifier | MLP, Linear | MLP |
| Optimizer | SGD, Adam | Adam |
| Batch Size | [32,64,128] | 32 |
| Embedded Vector's Size | [64,128,256,512] | 512 |

## A.2  Metrics

We evaluate the performance of our multi-tab WF attack using standard metrics from related studies [7, 15]. In the closed-world scenario, we treat multi-tab WF as a multi-class classification problem, using Top-k Accuracy (A@K), Precision, Recall, and F1-score

to comprehensively assess the model's performance across all websites. Let $y$ denote the true label vector for an instance $x$, where $y_i = 1$ indicates that $x$ visited the $i$-th website, and $y_i = 0$ otherwise. Similarly, $\hat{y}$ represents the predicted label vector, where each element corresponds to the predicted probability that $x$ visited a specific website. A@K is used to measure how many of them visited websites appear in the top-k predicted websites based on $\hat{y}$. The formulation of A@K is as follows:

$$A@k = \frac{1}{k} \sum_{l \in r_k(\hat{y})} y_l, \tag{1}$$

where $r_k(\hat{y})$ denotes the set of websites with the top-k highest probabilities in the predicted label vector $\hat{y}$. This metric evaluates the effectiveness of the model in ranking the true websites among the top predictions. Additionally, we compute Precision, Recall, and F1-score to assess the performance of our model further. These metrics are calculated based on the number of true positives (TP), false positives (FP), true negatives (TN), and false negatives (FN) for each website. The formulas for these metrics are given as: Precision = $\frac{TP}{TP+FP}$, Recall = $\frac{TP}{TP+FN}$, and F1-Score = $\frac{2 \times \text{Precision} \times \text{Recall}}{\text{Precision}+\text{Recall}}$. The average values of these metrics are computed across all websites to provide an overall performance measure.

In the open-world scenario, we evaluate the proposed method following established practices [20, 21]. Non-monitored websites are grouped into a single category, allowing us to test FMWF's ability to identify visited sites among numerous unseen ones. This approach assesses the model's adaptability and robustness in more realistic browsing conditions.

