# OpenReview forum: "Beyond Single Tabs: A Transformative Few-Shot Approach to Multi-Tab Website Fingerprinting Attacks"
_ACM.org/TheWebConf/2025/Conference — WWW 2025 Oral_

### Official Review · Reviewer_SL7b · 2024-11-25

**Novelty:** 4
**Technical Quality:** 4

**Review:**

This paper introduces a novel approach called Few-shot Multi-tab Website Fingerprinting (FMWF) to combat multi-tab website fingerprinting attacks. This approach aims to address some of the limitations of existing website fingerprinting techniques, especially those scenarios that assume users are browsing in a single tab. FMWF addresses the challenges of mixed, overlapping traffic traces generated by multi-tab browsing through two key innovations:

1. Advanced data augmentation techniques
2. Powerful fine-tuning algorithm based on transfer learning

The paper experiments with real traffic traces collected from three widely used browsers - Microsoft Edge, Google Chrome, and Tor Browser - demonstrate the superior performance of FMWF in both closed-world and open-world scenarios. In particular, in the open-world scenario, FMWF improves the accuracy by at least 12.3% compared to some existing methods.

**Quality:**
The paper presents a sophisticated approach to multi-tab website fingerprinting attacks, introducing Few-shot Multi-tab Website Fingerprinting (FMWF). The quality of the research is commendable, with a clear focus on addressing the practical challenges of existing WF attacks.

**Clarity:**
The paper is well-organized, with a logical flow from problem statement to solution and evaluation.

**Originality:**
The originality of the work is evident in its innovative approach to handling multi-tab traffic traces with minimal real-world data. The concept of pre-training with augmented traffic sequences is novel and addresses a gap in the field.

**Significance:**
The significance of this work is high, as it tackles a real-world issue with practical implications for user privacy. The ability to accurately identify website visits through encrypted traffic is a critical concern for internet security and privacy.

**Pros:**
- Pioneering approach to multi-tab website fingerprinting with fewer data requirements.
- Effectiveness demonstrated against state-of-the-art methods in various scenarios.
- Addresses the realistic and complex nature of multi-tab browsing.

**Cons:**
- The paper may overstate the generalizability of the model without sufficient evidence of its robustness across different network conditions and website structures.
- The potential for the model to be evaded by simple countermeasures, such as traffic shaping or timing perturbations, is not discussed.
- The paper does not fully explore the long-term sustainability of the approach as websites and user behaviors continue to evolve.
- The data augmentation technique may introduce bias, and the paper could provide more details on how this is mitigated.

**Questions:**

1. How well does the FMWF model generalize to unseen websites or new traffic patterns?

2. Will the performance of the model be affected as the number of monitored websites increases? Can it be extended to larger network environments?

3. Do the experiments in the article take into account all possible deviations? Are the experimental results representative of real-world performance?

4. How robust is the model mentioned in the article to actual situations such as changing network conditions and website content updates?

**Reviewer Confidence:**

3: The reviewer is confident but not certain that the evaluation is correct

**Scope:**

3: The work is somewhat relevant to the Web and to the track, and is of narrow interest to a sub-community

---

### Official Review · Reviewer_Dawe · 2024-12-01

**Novelty:** 5
**Technical Quality:** 4

**Review:**

### Strength:
1. This paper focuses on an interesting and important question, which has not been fully solved before.
2. This paper is well-structured and easy to follow. Related code is open-sourced.
3. A novel lightweight architecture FMWF and data augmentation approach are proposed, achieving performance comparable to state-of-the-art methods.

### Weakness:
1. Lacks a quantitative evaluation of its claimed core contributions.
2. A detailed description of the datasets used in the comparison experiments is missing.

### Detailed Comments：

In general, I think this paper focuses on an important question and also proposes some attempts to solve the Multi-tab Website Fingerprinting problem. However, during the experiment, I have the following questions, as listed in the **Questions** Section

**Questions:**

1. The claim that FMWF requires a smaller amount of training data and is more lightweight is its core contribution. However, the evaluation section does not provide any experiments or explanations regarding these two aspects. More quantitative experimental data is needed, such as how many times smaller the required data volume is compared to the state-of-the-art or how much the computational overhead is reduced.
2. The paper mentions that several state-of-the-art methods have limitations, such as a decline in fingerprinting performance over time. How does FMWF perform in this regard? Have you conducted experiments on this? The paper does not explain whether the data augmentation method you proposed can address this performance decline issue.
3. Building on the previous point, while you compared several state-of-the-art methods in your experiments, you did not specify the collection times of their training and test datasets. Without this information, it is impossible to conclude that your superior performance over these methods is due to improvements in the methodology rather than the degradation of SOTA methods' performance over time.
4. The data augmentation method proposed in the paper generates multi-tab traffic traces using the "zero time interval" type, where there is no overlapping between the traffic traces. And you mention that this overlapping is a major challenge for your compared SOTA methods. However, the closed-world datasets used in AQ I–AQ IV are all multi-label datasets you generated using the "zero time interval" type, and AQ V also lacks any discussion of overlapping data in the open-world datasets you use. How can you demonstrate that your method can address the challenges posed by overlapping traffic traces?
5. Why was there no comparison with SOTA methods for AQ II and AQ III, while comparisons were conducted for the other AQs?
6. Here are some writing and formatting suggestions:
  1. On page 4, in the right column, line 463, there is an extra comma in the reference; it is recommended to remove it.
  2. In Figure 5, the y-axis title of the second subplot is incorrect; please modify it.
  3. In Section 4.2, there are two instances of AQ1; it is suggested to modify them to AQ1-1 and AQ1-2.

**Reviewer Confidence:**

3: The reviewer is confident but not certain that the evaluation is correct

**Scope:**

4: The work is relevant to the Web and to the track, and is of broad interest to the community

---

### Official Review · Reviewer_iD8D · 2024-12-01

**Novelty:** 5
**Technical Quality:** 4

**Review:**

This paper proposes Few-shot Multi-tab Website Fingerprinting (FMWF), a novel framework addressing the challenges of multi-tab website fingerprinting attacks. The authors effectively tackle limitations such as the need for large-scale datasets and frequent retraining due to evolving website content.

Strengths:
1. The paper introduces a well-defined, two-stage framework addressing critical gaps in multi-tab WF attacks.
2. The experiments span multiple browsers (Tor, Chrome, Edge) and scenarios (closed-world, open-world).
3. Minimal real-world data requirements make the approach scalable and practical for real-world applications.
4. Robust accuracy improvements and reduced training costs make FMWF highly significant.

Weaknesses:
1. While the method is tested on multiple datasets, the effectiveness on other types of traffic (e.g., highly obfuscated traffic) could be explored further.
2. The adaptability of FMWF to environments with rapidly evolving user behaviors is discussed but could benefit from deeper investigation.
3. Although zero-time intervals (when users load multiple websites simultaneously) are mentioned, they are not adequately analyzed in experiments. This limits the framework's applicability in such real-world scenarios.
4. This paper presents strong comparative results but does not delve into the reasons for performance variations across different models.

**Questions:**

1. How does FMWF handle scenarios where significant variability exists in traffic patterns, such as websites employing advanced traffic obfuscation techniques?
2. How does the choice of augmented data (e.g., synthesized combinations of tabs) influence model performance compared to real-world multi-tab traces?
3. Can you elaborate on the potential applications of FMWF in ethical scenarios, such as detecting malicious traffic patterns, and how to mitigate misuse?
4. Why were zero-time interval scenarios not included in the experimental analysis? These scenarios seem to have practical significance.
5. Have other classification algorithms (e.g., transformers, gradient boosting) been evaluated for their effectiveness in FMWF?

**Reviewer Confidence:**

2: The reviewer is willing to defend the evaluation, but it is likely that the reviewer did not understand parts of the paper

**Scope:**

4: The work is relevant to the Web and to the track, and is of broad interest to the community

---

### Official Review · Reviewer_fL63 · 2024-12-02

**Novelty:** 6
**Technical Quality:** 3

**Review:**

The authors have designed and evaluated a new ML model to address multi-tab website fingerprinting using few-shot learning. This model is a step toward practical website fingerprinting.

The paper is well written in general, but there are some redundancies, and it is missing some crucial details.  The link provided to the code and data shows a 404 error.

The paper missed a clear and exact description of the NN and MLP model architectures. It also missed an explanation of what else the authors might have tried and tested and more details on their design choices.

Some of the evaluation metrics also leave some questions for the reader.

**Questions:**

You explain that for data augmentation, you used zero time interval type. Wouldn’t it be more realistic to use all three types? How would your model perform if you would have used all three types?

Most people I know have 10-20 or more tabs open in multiple browsers. Although not all tabs are usually actively used, I am curious if your model would be practical in these scenarios.

For Table 4, why did you use A@x for x-tab? Why didn’t you use the same metric for different tab numbers, making the evaluation comparable?

For Table 5, why you didn’t show results for some other metrics? Precision, recall, etc? A@2 is a bit like cheating because even if your classifier finds the unmonitored class often, it might look okay with this metric.

How did you choose the  unmonitored URLs? Using top websites might not be interesting for real-life scenarios (monitored and unmonitored).

How would your system have performed if you had used 100,000 unmonitored traces? That would have made me more confident that the system would have worked on many different users browsing different websites.

**Reviewer Confidence:**

3: The reviewer is confident but not certain that the evaluation is correct

**Scope:**

4: The work is relevant to the Web and to the track, and is of broad interest to the community